# C$^3$-Bench: Evaluating and Achieving Controllable Code Completion in Code LLM

## Abstract

Code completion has become a central task, gaining significant attention with the rise of large language model (LLM)-based tools in software engineering. Although recent advances have greatly improved LLMs' code completion abilities, evaluation methods have not advanced equally. Most current benchmarks focus solely on functional correctness of code completions based on given context, overlooking models' ability to follow user instructions during completion—a common scenario in LLM-assisted programming. To address this limitation, we present the first instruction-guided code completion benchmark, **C**ontrollable **C**ode **C**ompletion Benchmark (C$^3$-Bench), comprising 2,195 carefully designed completion tasks. Through comprehensive evaluation of over 40 mainstream LLMs across C$^3$-Bench and conventional benchmarks, we reveal substantial gaps in instruction-following capabilities between open-source and advanced proprietary models during code completion tasks. Moreover, we develop a straightforward data synthesis pipeline that leverages Qwen2.5-Coder to generate high-quality instruction-completion pairs for supervised fine-tuning (SFT). The resulting model, Qwen2.5-Coder-C$^3$, achieves state-of-the-art performance on C$^3$-Bench. We further investigate the interplay between instruction-following and code completion correctness, finding that performance on C$^3$-Bench strongly correlates with results from coding arenas. All code and datasets are available at https://anonymous.4open.science/r/Controllable-Code-Completion-Benchmark-42A3.

## 1 Introduction

Code completion represents a specialized code generation task that requires models to generate intermediate code segments while considering both left and right context (Bavarian et al., 2022; Allal et al., 2023). Recent advances in commercial foundation models, including GPT series (OpenAI, 2023), Claude series (Anthropic, 2023a), and Gemini series, have demonstrated remarkable capabilities in code generation tasks. Concurrently, open-source code LLMs such as StarCoder (Lozhkov et al., 2024), DeepSeekCoder (Guo et al., 2024), and Qwen-Coder (Hui et al., 2024) have achieved competitive performance compared to leading proprietary LLMs in code completion tasks. These advancements have facilitated the emergence of numerous LLM-powered code applications, including *GitHub Copilot*[1], *Cursor*[2], and *Devin*[3], which are significantly enhancing developers' productivity throughout the software development lifecycle.

When utilizing LLM-powered code applications like *Cursor Composer* and *Copilot Chat*, developers frequently need models not only to generate middle code based on context but also to follow specific implementation instructions. However, traditional benchmarks such as HumanEval (Chen et al., 2021a), CrossCodeEval (Ding et al., 2023), and SAFIM (Gong et al., 2024) provide limited evaluation of code completion capabilities, focusing solely on functional correctness through similarity metrics or unit tests while overlooking models' instruction-following abilities. With the increasing adoption of LLM-based code completion tools in software development, the ability to follow user-specified instructions has become increasingly critical for practical applications. There is thus a pressing need for new evaluation methodologies that can effectively assess models' ability to generate code

---

[1] https://github.com/features/copilot
[2] https://www.cursor.com
[3] https://devin.ai

completions following user-specified fine-grained instructions, providing a more comprehensive evaluation of code completion capabilities in practical development scenarios.

To effectively evaluate models' instruction-following capabilities in code completion tasks, we propose the concept of Controllable Code Completion (CCC). As illustrated in Figure 1, CCC extends traditional code completion by incorporating diverse middle code variants and fine-grained control instructions. This enhancement enables comprehensive assessment of both functional correctness and instruction adherence, providing a more complete evaluation of code completion capabilities. A detailed example is presented in Figure 7. Building upon this concept, we introduce $C^3$-Bench (Controllable Code Completion benchmark), comprising 2,195 high-quality, instruction-guided test cases. The benchmark implements two primary evaluation mechanisms: **Implementation-Control Completion (ICC)** evaluates models' ability to follow specific implementation requirements.

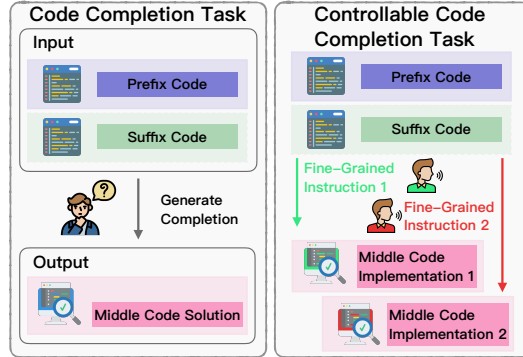

Figure 1: Comparison between Controllable Code Completion and traditional code completion tasks. The former extends standard code context by incorporating fine-grained instructions to guide the completion process.

Test cases share identical code context but vary in implementation instructions, covering four categories: *Structural Specification*, *Algorithmic Implementation*, *Control Flow*, and *Critical Parameter Requirements*. **Scale-Control Completion (SCC)** assesses models' ability to generate code of specified scope, including *Line Span*, *Multi-line*, and *Statement Block* completions. Notably, $C^3$-Bench employs automated scoring mechanisms, ensuring objective evaluation without human intervention.

We conduct comprehensive evaluations of over 40 mainstream general-purpose LLMs and code LLMs on both $C^3$-Bench and conventional code completion benchmarks, providing detailed cross-benchmark performance analysis. The experimental results reveal widespread limitations in instruction-following capabilities among LLMs, suggesting that their code completion capabilities in real-world development scenarios may not match their performance on existing benchmarks. Moreover, while open-source code LLMs achieve competitive performance with proprietary LLMs on conventional benchmarks, $C^3$-Bench reveals a substantial instruction-following performance gap between them, indicating that open-source code models may overfit to existing benchmarks and lack sufficient generalization capabilities in code completion tasks. Furthermore, performance on $C^3$-Bench strongly correlates with results from the Copilot Arena (Chi et al., 2025), underscoring its practical relevance. To enhance models' instruction-following capabilities in code completion, we propose an automated training data synthesis pipeline. This pipeline leverages Qwen2.5-Coder-32B-Instruct to generate large-scale instruction-completion pairs from unsupervised GitHub repository code data (Lozhkov et al., 2024). Utilizing these synthesized training data, we develop Qwen2.5-Coder-$C^3$, which achieves state-of-the-art performance in controllable code completion while maintaining its competence on conventional code completion benchmarks.

Our contributions are summarized as follows:
- We identify the limitations of existing benchmarks in comprehensively evaluating code completion abilities and present the first instruction-guided benchmark, **C**ontrollable **C**ode **C**ompletion Benchmark, to assess both functional correctness and instruction-following capabilities during code completion.
- We present the first comprehensive assessment of code completion capabilities, evaluating over 40 general-purpose and code-specialized LLMs across multiple benchmarks. Our analysis reveals that current evaluation methods systematically overestimate models' capabilities in practical applications and identifies significant performance gaps between open-source and proprietary models. These findings provide valuable insights and directions for future research in enhancing code completion capabilities of language models.
- We develop a straightforward pipeline for synthesizing instruction-completion training pairs and leverage these for supervised fine-tuning, producing Qwen2.5-Coder-$C^3$ with enhanced instruction-following capabilities in code completion tasks, contributing to the advancement of open-source code LLMs.

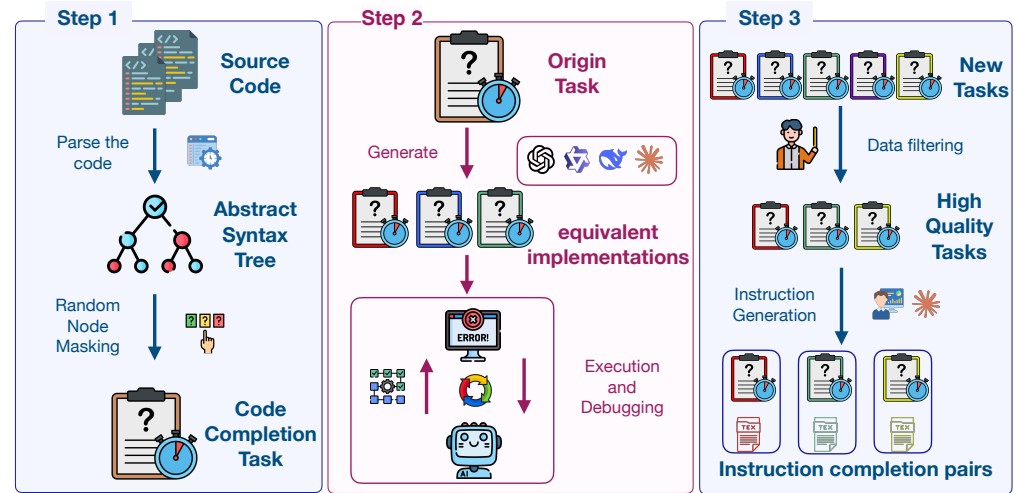

Figure 2: Overview of the construction pipeline of C³-Bench.

## 2 C³ BENCHMARK

In this section, we introduce the overview of **C**ontrollable **C**ode **C**ompletion Benchmark, with considerations of definition (Section 2.1), datasets (Section 2.2) construction (Section 2.3) and evaluations (Section 2.4).

### 2.1 DEFINITION OF CONTROLLABLE CODE COMPLETION

**Controllable Code Completion (C³)** extends the traditional code completion paradigm. A conventional code completion instance is defined as a tuple $(P, S, G, T)$, where $P$ (**Prefix Code**) denotes left code context, $S$ (**Suffix Code**) represents right code context, $G$ (**Ground-Truth Middle Code**) indicates the expected middle code implementation, and $T$ (**Unit Test**) comprises test cases validating $G$. A CCC instance augments this framework by incorporating $I$ (**Fine-Grained Instruction**), which specifies implementation requirements, thus forming a tuple $(P, S, G, T, I)$. Given a dataset $\{(P_i, S_i, G_i, T_i, I_i)\}$, we train an LLM $M$ to generate completions such that $M(P_i, S_i, I_i) \rightarrow G_i$. The evaluation encompasses two aspects: functional correctness, verified through $T_i$, and instruction adherence, assessed by measuring the alignment between the implementation approach in $G_i$ and the requirements specified in $I_i$. Based on the nature of instruction $I$, we categorize CCC tasks into two distinct types: **Implementation-Control Completion (ICC)** and **Scale-Control Completion (SCC)**.

**Definition 2.1.** ***Implementation-Control Completion (ICC)*** specifies detailed requirements for middle code implementation, demanding models to generate complete and functionally correct code that passes unit tests. The implementation requirements are categorized into four primary types: ***1. Structural Specification Requirements:*** Code organization and architecture specifications including basic data structure definitions, composite data type design, class/interface structure specifications, and data model design requirements. ***2. Algorithmic Implementation Requirements:*** Specific algorithmic approaches encompassing core algorithm flow, computational logic implementation, data transformation processing, and optimization strategy requirements. ***3. Control Flow Requirements:*** Program execution patterns involving execution flow definition, branch logic handling, iteration structure design, and exception handling mechanisms. ***4. Critical Parameter Requirements:*** Parameter and variable management including core variable definition specifications, parameter passing rules, state variable management, and configuration parameter settings.

**Definition 2.2.** ***Scale-Control Completion (SCC)*** implements fine-grained control over the scope of middle code completion, wherein models are required to generate code segments of precisely specified scale rather than complete functional implementations. Given its focus on structural conformity rather than functional completeness, this category does not employ unit test validation. The scale requirements are systematically categorized into three distinct types: ***1. Line Span Completion:*** pertains to the completion of partial code lines; ***2. Multi-line Completion:*** mandates the generation of a predetermined number of complete code lines; and ***3. Statement Block Completion:***

encompasses the completion of specific control structures, including `IF STATEMENT BLOCK`, `FOR STATEMENT BLOCK`, and `WHILE STATEMENT BLOCK`.

Table 1: Statistical analysis of $C^3$-Bench dataset. Token counts (min/max/mean) are reported for each component across ICC and SCC tasks.

| | **I**mplementation **C**ontrol **C**ompletion | | | | |
| | Structural | Algorithmic | Control-Flow | Parameter | Average |
|---|---|---|---|---|---|
| \|Samples\| | 111 | 502 | 547 | 126 | - |
| Instruction Tokens | 4/20/11 | 4/27/10 | 4/26/10 | 5/20/10 | 4/27/10 |
| Prefix Tokens | 50/1072/548 | 27/1447/413 | 38/1447/367 | 38/1252/455 | 27/1447/413 |
| Middle Tokens | 6/310/87 | 6/371/75 | 10/709/72 | 5/262/64 | 5/709/73 |
| Suffix Tokens | 1/529/31 | 1/615/55 | 1/1455/86 | 1/1132/154 | 1/1455/57 |

| | **S**cale **C**ontrol **C**ompletion | | | | |
| | Span | Multi-Lines | Statement-Block | | Average |
|---|---|---|---|---|---|
| \|Samples\| | 97 | 467 | 345 | - | - |
| Instruction Tokens | 6/10/8 | 6/11/8 | 6/11/8 | - | 6/11/8 |
| Prefix Tokens | 342/1224/623 | 257/1593/654 | 133/2717/665 - | - | 133/2717/655 |
| Middle Tokens | 2/82/9 | 8/1083/102 | 2/192/34 | - | 2/1083/66 |
| Suffix Tokens | 5/93/37 | 3/211/46 | 3/1825/123 | - | 3/1825/77 |

## 2.2 DATASET STATISTICS

We present comprehensive dataset statistics in Table 1. $C^3$-Bench comprises 2,195 high-quality Python CCC instances, encompassing 1,286 ICC and 909 SCC task instances, respectively. All test cases within the ICC task are accompanied by corresponding unit tests. The dataset and its associated unit tests are derived from two widely-used, high-quality code evaluation datasets: HumanEval (Chen et al., 2021a) and SAFIM (Gong et al., 2024). To enhance task complexity and diversity, we have extracted extended middle code segments and developed multiple implementation variants, each accompanied by carefully crafted detailed instructions. These enhancements facilitate a more rigorous evaluation of LLMs' capabilities in following diverse implementation requirements while maintaining functional correctness.

## 2.3 BENCHMARK CONSTRUCTION

Figure 2 illustrates the synthesis pipeline for constructing $C^3$-Bench, which consists of four main steps: (1) middle code extraction (2) equivalent implementation generation (3)data filtering and instruction generation, which are described in detail below.

### 2.3.1 MIDDLE CODE EXTRACTION

The original HumanEval and SAFIM datasets primarily contain single-line implementations as ground truth middle code, which limits the complexity and scope for instruction-guided completion. To address this limitation, we develop a systematic extraction approach utilizing Abstract Syntax Trees (AST). ASTs represent Python code as hierarchical tree structures, with each node corresponding to a specific code construct and capturing syntactic nesting relationships. Leveraging `tree-sitter-languages` [4], we parse code snippets and extract logically complete code blocks that maintain semantic coherence, a crucial requirement for meaningful instruction-guided completion. Our extraction process comprises two steps: (1) Systematic traversal and manipulation of ASTs, masking nodes at multiple levels to generate new middle code segments; (2) Additional masking of 3-5 consecutive code lines for 30% of the instances, specifically designated for SCC tasks.

### 2.3.2 EQUIVALENT IMPLEMENTATION GENERATION

For ICC tasks, we manually authored functionally equivalent implementations for the extracted middle code segments, while SCC tasks directly utilize the segments from the previous step. We assembled

---

[4]https://pypi.org/project/tree-sitter-languages/

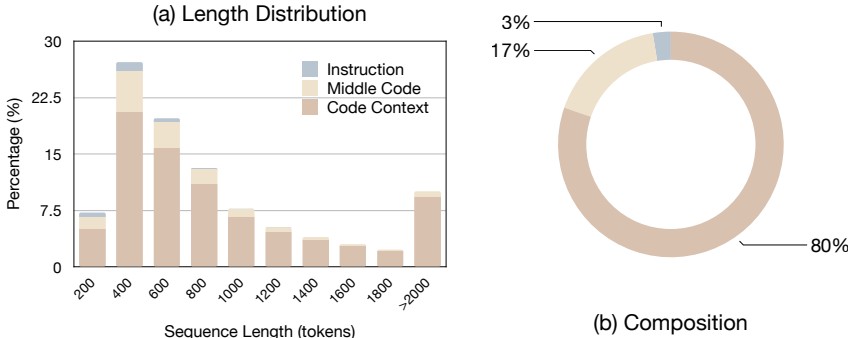

Figure 3: Training dataset analysis showing (a) sequence length distribution and (b) average component composition (Context Code, Middle Code, and Instruction).

a team of three senior Python developers who systematically rewrote the middle code segments extracted in the previous step to produce synonymous implementations. To ensure high quality and correctness, the developers followed a rigorous authoring process, including cross-validation and peer reviews, to verify that each new implementation was functionally identical to the original. This meticulous process yielded over 6,000 distinct implementations, with more than 50% of cases featuring at least three unique implementation variants for the same code context. Detailed examples of these variants are presented in Appendix F.

### 2.3.3 DATA FILTERING AND INSTRUCTION GENERATION

In the final stage of our pipeline, we implement a rigorous quality control process to ensure benchmark reliability. Building upon instances that passed unit testing in the previous step, we apply comprehensive filtering criteria: code readability adhering to PEP8 standards, appropriate length constraints (middle code $\leq$ 30% of total context), significant implementation diversity, and algorithmic efficiency. For filtered instances, we employ distinct instruction generation approaches: ICC tasks receive manually crafted implementation specifications, while SCC tasks utilize Claude3.5-Sonnet-generated scope requirements. All instructions maintain precise expression and task-specific focus (implementation methodology for ICC, scale specifications for SCC). The instruction quality undergoes systematic validation through both expert review (five senior Python developers) and automated consistency checking (Claude3.5-Sonnet), ensuring reliable assessment of models' code understanding capabilities. Detailed examples are presented in Appendix F.

### 2.4 EVALUATION METRICS

To accurately assess model performance on C$^3$-Bench, we employ three complementary metrics across Implementation-Control Completion (ICC) and Scale-Control Completion (SCC) tasks: (1) **Pass@1** evaluates functional correctness in ICC tasks through unit testing (Chen et al., 2021b); (2) **Instruction-Following Rate (IF)** measures adherence to specified requirements, assessed only for functionally correct cases in ICC tasks; and (3) **Edit Similarity (ES)** serves as a supplementary static analysis metric for preliminary validation of generation quality. The IF evaluation implements task-specific approaches: **Semantic Validation for ICC:** We employ a LLM-based judging system with Claude3.5-Sonnet as the primary judge (system prompt in Figure 8). The system's reliability is validated through extensive experiments, achieving 98% agreement with senior Python developers across 10 independent assessment rounds. Additionally, we provide Qwen2.5-32B-Instruct as a cost-effective alternative for the research community. **Structural Verification for SCC:** We implement two automated approaches: (1) AST-based node type matching for structural requirements and (2) length-based verification for line count specifications, both leveraging `tree-sitter-languages` for systematic code analysis.

## 3 QWEN2.5-CODER-C$^3$

### 3.1 DATA SYNTHESIS

To address the scarcity of instruction-completion training pairs, we propose a straightforward automated synthesis pipeline utilizing Python code from GitHub repositories (Lozhkov et al., 2024).

| | CCC-Bench | CCEval | RepoEval | CCLongEval | ExecRepoBench | SAFIM | Copilot Arena (Coding) |
|---|---|---|---|---|---|---|---|
| **CCC-32B** | 1 | -1 | -1 | -4 | -5 | -3 | - |
| o1-2024-12-17 | 2 | -7 | -2 | -5 | 0 | -3 | +1 |
| Claude3.5-Sonnet | 3 | -2 | 0 | 0 | +2 | +1 | +2 |
| GPT-4o-1120 | 4 | -4 | -5 | -5 | 0 | -5 | -1 |
| DeepSeek-V3 | 5 | -2 | -2 | -3 | 0 | -2 | +2 |
| Gemini-2.0-Flash | 5 | -1 | 0 | -1 | +3 | -1 | +2 |
| Qwen2.5-Coder-32B | 7 | +6 | +6 | +6 | -1 | +6 | +1 |
| Codestral-22B-V0.1 | 8 | +5 | 0 | +6 | -1 | 0 | - |
| DS-Coder-33B | 9 | +5 | +3 | +5 | +2 | +6 | +2 |
| | | r=-0.28 | r=0.40 | r=-0.38 | r=0.60 | r=-0.04 | r=0.92 |

Figure 5: Cross-benchmark comparison of model rankings. Base rankings from $C^3$-Bench (leftmost column) are compared with relative ranking changes in existing benchmarks, indicated by color-coded values (green: performance improvement, red: performance degradation, dash: model not evaluated). Spearman correlation coefficients ($r$) quantify ranking consistency. **CCC-32B** represents Qwen2.5-Coder-32B-$C^3$.

Our pipeline follows a two-phase bootstrapping approach: (1)**Initial Seed Generation:** We leverage Claude3.5-Sonnet to generate 1,000 high-quality instruction-completion pairs through middle code extraction and instruction generation. These pairs serve as seed examples to guide subsequent large-scale synthesis. (2)**Automated Synthesis:** Using the seed examples as few-shot demonstrations, we employ Qwen2.5-Coder-32B-Instruct for automated middle code extraction and instruction generation. The extracted middle code segments undergo validation through pattern matching to ensure accuracy. This simple yet effective approach enables the generation of large-scale, high-quality $C^3$ task training data. Detailed statistic analysis of the synthesized data are presented in Figure 3.

## 3.2 MODEL TRAINING

We develop Qwen2.5-Coder-$C^3$ by fine-tuning both Qwen2.5-Coder-1.5B and Qwen2.5-Coder-32B variants on 200,000 synthetic instruction-completion pairs generated from GitHub data (Lozhkov et al., 2024) using Qwen2.5-Coder-32B-Instruct. To ensure evaluation integrity, we perform 10-gram decontamination between the training data and $C^3$-Bench. The training process, implemented on 64 NVIDIA A100-80GB GPUs, employs the Adam optimizer (Kingma & Ba, 2015) with a learning rate of $3 \times 10^{-5}$ (50 warmup steps), a global batch size of 1024 samples, tensor parallel size of 2, and 4K tokens sequence truncation.

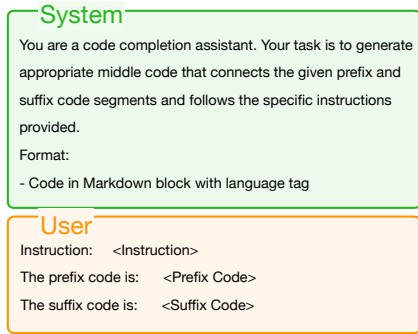

Figure 4: The ChatML format of $C^3$-bench. <Instruction > indicts the detail control information of completion. <Prefix Code > and <Suffix Code> are the prefix and suffix code.

Table 2: Performance comparison of different LLMs on C³-Bench. **Bold** and underlined values denote the best and second-best performance metrics respectively within the same model size range. The Average column represents the mean IF rate across tasks. **FIM** column indicates whether models were evaluated using Fill-In-the-Middle special token format (✓) or not (✗).

| Size | Model | FIM | Implementation-Control | | | Scale-Control | | Average |
|------|-------|-----|-----|--------|-----|-----|-----|-----|
| | | | *ES* | *pass@1* | *IF* | *ES* | *IF* | *IF* |
| 1B+ Models | DeepSeek-Coder-1.3B-Instruct | ✗ | 24.4 | 30.7 | 6.1 | 14.3 | 5.0 | 5.5 |
| | DeepSeek-Coder-1.3B-Base | ✓ | 35.6 | 26.9 | 14.2 | 26.5 | 4.4 | 9.3 |
| | Qwen2.5-Coder-3B-Instruct | ✗ | 43.0 | 40.5 | **29.7** | 26.6 | 14.2 | 21.9 |
| | Qwen2.5-Coder-3B | ✓ | 42.3 | **44.8** | 26.3 | 30.1 | 10.1 | 18.2 |
| | Qwen2.5-Coder-1.5B | ✓ | 38.3 | 34.0 | 19.0 | 26.4 | 3.6 | 11.3 |
| | Qwen2.5-Coder-1.5B-Instruct | ✗ | 12.4 | 22.9 | 0.7 | 9.2 | 8.0 | 4.3 |
| | **Qwen2.5-Coder-1.5B-C³** | ✗ | 44.7 ↑32.3 | 39.7 ↑16.8 | 29.6 ↑28.9 | 40.4 ↑31.2 | **66.8 ↑58.8** | **48.2 ↑43.9** |
| 6B+ Models | DeepSeek-Coder-6.7B-Instruct | ✗ | 28.2 | 39.5 | 8.0 | 17.6 | 3.2 | 5.6 |
| | DeepSeek-Coder-6.7B-Base | ✓ | 40.7 | 41.8 | 27.1 | 29.9 | 4.9 | 18.2 |
| | DeepSeek-Coder-V2-Lite-Instruct | ✗ | 24.3 | 41.0 | 8.7 | 13.5 | 3.0 | 5.8 |
| | DeepSeek-Coder-V2-Lite-Base | ✓ | 40.9 | 43.7 | 27.5 | 28.9 | 4.1 | 15.8 |
| | Qwen2.5-Coder-7B-Instruct | ✗ | 37.3 | 44.2 | 21.9 | 19.2 | 5.0 | 13.4 |
| | Qwen2.5-Coder-7B | ✓ | 42.1 | **45.3** | **29.1** | 29.9 | **7.5** | **18.3** |
| | OpenCoder-8B-Instruct | ✗ | 19.5 | 35.5 | 1.6 | 12.1 | 2.7 | 2.1 |
| | Yi-Coder-9B-Chat | ✗ | 31.6 | 42.3 | 25.1 | 11.8 | 1.8 | 13.4 |
| 14B+ Models | StarCoder2-15B-Instruct-v0.1 | ✗ | 29.2 | 36.6 | 4.2 | 13.7 | 1.6 | 2.6 |
| | StarCoder2-15B | ✓ | 9.1 | 0.2 | 0.1 | 7.9 | 1.0 | 0.5 |
| | Qwen2.5-Coder-14B-Instruct | ✗ | 31.9 | 52.0 | 25.7 | 21.6 | **13.5** | 19.6 |
| | Qwen2.5-Coder-14B | ✓ | 45.8 | **56.1** | **36.2** | 30.3 | 8.7 | **22.5** |
| | CodeStral-22B-v0.1 | ✗ | 41.7 | 50.5 | 34.1 | 22.2 | 6.2 | 20.1 |
| 20B+ Models | DeepSeek-Coder-33B-Instruct | ✗ | 30.7 | 41.6 | 15.0 | 18.7 | 4.7 | 9.9 |
| | DeepSeek-Coder-33B-Base | ✓ | 40.7 | 48.1 | 32.0 | 29.3 | 5.2 | 18.6 |
| | CodeLlama-34B-Instruct | ✗ | 23.7 | 12.7 | 3.4 | 16.9 | 5.0 | 4.2 |
| | CodeLlama-70B-Instruct | ✗ | 31.9 | 32.0 | 14.3 | 13.9 | 4.6 | 9.5 |
| | Qwen2.5-72B-Instruct | ✗ | 23.3 | 47.0 | 9.8 | 21.8 | 9.4 | 9.6 |
| | DeepSeek-V3 | ✗ | 34.2 | 61.7 | 47.3 | 24.4 | 20.2 | 33.8 |
| | DeepSeek-V3-0324 | ✗ | 29.5 | 59.4 | **53.0** | 24.8 | 20.2 | 36.6 |
| | Qwen2.5-Coder-32B | ✓ | 46.7 | 58.1 | 38.7 | 30.9 | 5.2 | 21.9 |
| | Qwen2.5-Coder-32B-Instruct | ✗ | 30.2 | 49.8 | 28.8 | 20.9 | 16.9 | 22.8 |
| | **Qwen2.5-Coder-32B-C³** | ✗ | 49.3 ↑19.1 | 62.0 ↑12.2 | 52.5 ↑23.7 | 44.2 ↑23.3 | **80.7 ↑63.8** | 66.6 ↑43.8 |
| Closed-APIs | GPT-4o-mini-2024-07-18 | ✗ | 39.9 | 49.1 | 42.2 | 30.1 | 22.5 | 32.9 |
| | GPT-4-2024-06-13 | ✗ | 44.0 | 58.9 | 48.3 | 32.2 | 12.5 | 30.4 |
| | GPT-4o-2024-05-13 | ✗ | 39.1 | 54.3 | 41.3 | 34.3 | 20.8 | 31.0 |
| | GPT-4o-2024-08-06 | ✗ | 39.3 | 65.8 | 58.6 | 35.0 | 24.1 | 40.8 |
| | GPT-4o-2024-11-20 | ✗ | 36.3 | 65.9 | 59.6 | 33.3 | 24.1 | 41.9 |
| | o1-mini | ✗ | 47.0 | 64.7 | 55.0 | 31.7 | 44.7 | 49.8 |
| | o1-preview | ✗ | 45.1 | 70.1 | 57.7 | 32.2 | 48.9 | 53.0 |
| | o1-2024-12-17 | ✗ | 31.3 | 72.1 | 62.9 | 36.3 | 59.6 | 61.3 |
| | Claude3.5-Haiku | ✗ | 29.7 | 54.2 | 40.7 | 27.2 | 26.4 | 33.6 |
| | Claude3.5-Sonnet-20241022 | ✗ | 30.2 | 68.8 | 60.9 | 32.3 | 50.8 | 55.8 |
| | Gemini-1.5-Pro-Flash | ✗ | 45.9 | 60.1 | 41.8 | 33.1 | 9.3 | 25.5 |
| | Gemini-2.0-Flash | ✗ | 36.9 | 70.7 | 59.5 | 28.9 | 7.0 | 33.2 |

# 4 BENCHMARKING STATE-OF-THE-ART MODELS

## 4.1 PROMPT FORMAT

For all experiments in this work, we employ two distinct prompting strategies based on model capabilities. For models supporting Fill-In-the-Middle (FIM) format (e.g., DeepSeek and Qwen), we utilize special token prompts as described in Hui et al. (2024). For other models, primarily chat-oriented ones, we employ the ChatML-formatted (OpenAI, 2022) prompt template illustrated in Figure 4, which explicitly specifies input requirements and expected output formats.

## 4.2 EXPERIMENTAL SETUP

We conduct comprehensive evaluations across 40+ models spanning diverse parameter scales, encompassing both general-purpose and code-specialized LLMs from open and proprietary sources. The evaluated models include: **General-purpose LLMs**: GPT series (OpenAI, 2023), Claude series (Anthropic, 2023b), Gemini series (Team & etc., 2024), Qwen2.5-72B-Instruct (team & etc., 2025), DeepSeek-V3 (DeepSeek-AI & etc., 2024), and o1-series. **Code-specialized LLMs**: CodeLlama

(Rozière et al., 2023), Qwen-Coder (Hui et al., 2024), DeepSeek-Coder (Guo et al., 2024), StarCoder (Lozhkov et al., 2024), Yi-Coder (01.AI, 2024), Codestral (MistralAI, 2024), and OpenCoder (Huang et al., 2024). We evaluate these models on multiple benchmarks: $C^3$-Bench, CrossCodeEval (Ding et al., 2023)(CCEval), RepoEval (Zhang et al., 2023), CrossCodeLongEval (Wu et al., 2024)(CC-LongEval), ExecRepoBench (Yang et al., 2024), and SAFIM (Gong et al., 2024). Notably, this work presents the first comprehensive assessment of advanced general-purpose models' code completion capabilities. For implementation, we employ vllm (Kwon et al., 2023) for open-source model inference, using greedy sampling with a 1024 tokens length limit. For chat models, we extract code from markdown blocks for evaluation.

## 4.3 PERFORMANCE ANALYSIS

We present comprehensive evaluation results through multiple perspectives: Table 2 shows detailed metrics on $C^3$-Bench, Figure 5 illustrates cross-benchmark ranking comparisons, and Appendix D provides complete benchmark results in Tables 3, 4, 5, 6 and Figure 10. Notably, Qwen2.5-Coder-32B-$C^3$ achieves state-of-the-art performance on $C^3$-Bench, demonstrating substantial improvements in instruction-following capabilities compared to Qwen2.5-Coder-32B-Instruct while maintaining competitive performance across other benchmarks.Our analysis reveals several key findings:

**Gap in Instruction Following:** While lightweight open-source code LLMs outperform proprietary models on similarity-based benchmarks (e.g., CrossCodeEval, RepoEval), they show significant limitations in instruction-following capabilities on $C^3$-Bench. This gap suggests potential challenges in meeting real-world development requirements where specific implementation guidance is crucial.
**Performance Variations Across Instruction Types:** Figure 12 demonstrates how models respond differently to implementation and scale-control instructions. Despite similar capabilities in following implementation guidelines (e.g., Gemini-2.0-Flash and o1), models exhibit substantial variations in scale-controlled code completion. Advanced LLMs including Gemini, DeepSeek-V3, and GPT-4o series struggle with scale-control tasks, indicating potential limitations in their training objectives.

**Correlation with Advanced Capabilities:** Model rankings on $C^3$-Bench show strong correlation with performance on tasks requiring extensive context understanding (ExecRepoBench) and user experience evaluation (Chatbot Arena-Coding (Berkeley et al., 2024)). This alignment suggests a potential relationship between instruction-following ability and broader code comprehension capabilities, offering directions for future research.

**Effectiveness of Instruction Tuning:** Our synthetic training data significantly improves models' instruction-following capabilities. While Qwen2.5-Coder-$C^3$ achieves superior performance in SCC tasks, surpassing proprietary LLMs, its ICC performance remains limited by the base model's capabilities, particularly in achieving high Pass@1 rates. These results provide valuable insights for enhancing instruction-following capabilities in open-source code LLMs.

## 4.4 ABLATION STUDY OF FINE-GRAINED INSTRUCTIONS

In this section, we examine the effectiveness of instructions in $C^3$-Bench through an ablation study on ICC tasks. We evaluate five representative models: o1-preview, Claude3.5-Sonnet-1022, DeepSeek-V3, Qwen2.5-Coder-32B-Instruct, and Qwen2.5-Coder-32B-$C^3$, comparing their performance with and without instruction guidance. As shown in Figure 6, removing instructions from query prompts leads to significant degradation in *IF* while *Pass@1* remain largely unchanged, with some models (e.g., o1-preview) showing slight improvements. These results demonstrate the substantial guiding effect of fine-grained instructions in $C^3$-Bench on code completion tasks, while also validating our benchmark's capability to evaluate models' instruction-following abilities in code completion.

## 5 RELATED WORKS

**Code Large Language Model.** Recent years, Large Language Models (LLMs) have achieved unprecedented advances in coding capabilities. Leading proprietary LLMs like GPT (OpenAI, 2023) and Claude (Anthropic, 2023a) demonstrate exceptional code generation and understanding abilities across multiple programming tasks. Specialized code-centric LLMs (Scao et al., 2022; Li et al., 2022; Fried et al., 2022; Jiang et al., 2024; Nijkamp et al., 2023; Wei et al., 2023; Zhao et al., 2024), like CodeLlama (Rozière et al., 2023), DeepSeek-Coder (Guo et al., 2024), OpenCoder (Huang et al.,

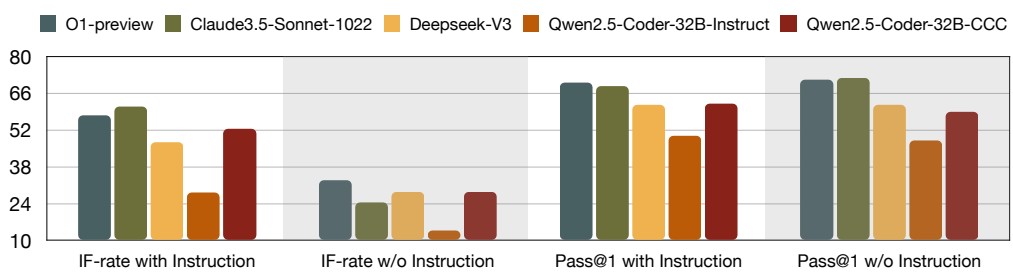

Figure 6: Impact of instruction guidance on model performance. Pass@1 and IF-rate metrics are compared across models under two conditions: with instructions and without instructions .

2024), and Qwen-Coder (Hui et al., 2024), excel in targeted tasks including code debugging (Huq et al., 2022), translation (Jiao et al., 2023), and completion (Bavarian et al., 2022). These models leverage domain-specific architectures and are all trained on vast corpuses comprising billions of code snippets to optimize programming-related performance. The evaluation landscape has evolved through comprehensive benchmarks assessing code quality. HumanEval (Chen et al., 2021a) and MBPP (Austin et al., 2021) provide foundational metrics, while EvalPlus (Liu et al., 2023) introduces enhanced testing protocols. Multilingual and multi-task frameworks including MultiPL-E (Cassano et al., 2023), McEval (Chai et al., 2024), MdEval (Liu et al., 2024b), and BigCodeBench (Zhuo et al., 2024) enable rigorous assessment across languages, paradigms, and task complexities.

**Code Completion.** Code completion tasks require models to generate missing code segments by leveraging both left and right contexts, providing crucial assistance for software development, several . Several benchmarks have been developed to evaluate models' code completion capabilities. HumanEval-FIM (Zheng et al., 2023), DS-1000 (Lai et al., 2023), and SAFIM (Gong et al., 2024) focus on in-file completion scenarios, while CrossCodeEval (Ding et al., 2023), RepoEval (Zhang et al., 2023), CrossCodeLongEval (Wu et al., 2024), and ExecRepoBench (Yang et al., 2024) assess cross-file completion abilities considering broader repository contexts and dependencies. However, existing benchmarks rely solely on execution-based metrics (e.g. Pass@k) or static analysis techniques (e.g., exact match (EM) and edit similarity (ES)) to evaluate completion correctness, overlooking the assessment of models' controllability in code completion tasks.

Additional discussion of related research in LLM instruction-following capabilities and human preference-based evaluation approaches is presented in detail in Appendix A.

## 6 CONCLUSION AND FUTURE DIRECTIONS

In this paper, we identify that conventional code completion evaluation metrics are incomplete, particularly in assessing models' instruction-following capabilities during code completion. To address this limitation, we introduce $C^3$-Bench, a fine-grained instruction-guided benchmark that enables comprehensive evaluation of models' code comprehension abilities. Our extensive evaluation encompasses over 40 mainstream LLMs across multiple code completion benchmarks, providing detailed performance analyses.

Our investigation yields several significant findings: **(i)** contemporary LLMs demonstrate notable limitations in instruction-following capabilities during code completion, particularly in adhering to code scale control instructions; **(ii)** while open-source code LLMs achieve comparable performance to closed-source models on functional correctness benchmarks, they exhibit substantial gaps in instruction-following capabilities; and **(iii)** our straightforward instruction-pair synthesis approach effectively enhances models' instruction-following abilities. This work contributes to advancing open-source model development and provides valuable insights for future research in code completion.

Notwithstanding these contributions, several critical challenges warrant further investigation: **Data Diversity:** While $C^3$-Bench currently focuses on in-file Python tasks, future work should explore multi-language scenarios and repository-level tasks with extended context. **Base Model Capabilities:** Our findings indicate that base model capabilities significantly constrain ICC task performance, suggesting an important direction for future research.

## 7  ETHICS STATEMENT

The data used in the $C^3$-Bench benchmark is sourced exclusively from public repositories that are governed by licenses permitting their use in software and research. Our contributions fully adhere to the terms of these licenses. We did not use any data beyond what is publicly available and downloadable from Github. Our work did not involve the participation of any human subjects; we did not use crowdsourcing or recruit any external human workers for any part of the $C^3$-Bench benchmark's creation. All work, including the environment configuration, data curation, synthetic data generation, and the writing of this paper, was conducted entirely by the author team.

## 8  REPRODUCIBILITY STATEMENT

To ensure the reproducibility of our work, we provide a complete codebase with detailed instructions for replicating the $C^3$-Bench benchmark results and the Qwen2.5-Coder-$C^3$ training process. The evaluation framework, data synthesis methodology, and training hyperparameters are detailed in Section 2.4 and Section 3.2. To further facilitate community engagement and standardized evaluation, we plan to release a PyPI package and host a public leaderboard for the benchmark.

## 9  LLM USAGE

The use of Large Language Models (LLMs) in this work was limited to providing minor assistance with the writing and editing of the manuscript.

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

APPENDIX

## A    ADDITIONAL RELATED WORK

**Instruction-Following Capabilities of LLMs.** Recent studies have extensively explored LLMs' instruction-following capabilities in code generation tasks. Yan et al. (2025) introduced CodeIF for evaluating instruction adherence across diverse coding scenarios, while H et al. (2022) and P et al. (2023) leveraged reinforcement learning to enhance code generation quality. Liu et al. (2024a) further contributed through Conifer, a dataset designed to improve complex instruction-following in LLMs. Despite these advances in general code generation instruction-following, the specific challenges of instruction-guided code completion remain largely unexplored, representing a significant gap in current research.

**Human Preference-Based Evaluation.** Recent advancements in arena-based frameworks have provided novel insights into LLM capabilities. Chi et al. (2024) evaluates models in dynamic multi-agent environments, while Berkeley et al. (2024) implements human preference-based pairwise comparisons for assessment, though concerns have been raised regarding data access inequality and potential training biases (Singh et al., 2025). In code generation specifically, Chi et al. (2025) evaluates LLMs in real-world scenarios, revealing significant disparities between traditional benchmark performance and practical effectiveness. While these approaches effectively capture user preferences and real-world coding capabilities, their reliance on online deployment and user interaction data limits widespread applicability, particularly for evaluating open-source models. This limitation underscores the need for lightweight, generalizable benchmarks that can robustly assess models' code context understanding and completion capabilities without requiring extensive online infrastructure.

## B    EXAMPLE OF CONTROLLABLE CODE COMPLETION TASK

This figure 7 demonstrates a Controllable Code Completion task focusing on the implementation of the Shortest Path Faster Algorithm (SPFA). The figure is structured in three main components: the initial code context, followed by two distinct fine-grained implementation instructions. The code context presents a partially implemented SPFA function framework, including memory allocation for essential data structures such as distance array, visit markers, and predecessor tracking. The function signature indicates its application to weighted directed graphs, with parameters for start and end vertices along with the graph structure. Two fine-grained instructions are provided, each specifying different optimization strategies for SPFA:

- The first instruction requires implementation of Small Label First (SLF) optimization utilizing a deque data structure. This approach prioritizes vertices with smaller distance values by inserting them at the front of the deque, while vertices with larger distance values are appended to the back.
- The second instruction, accompanied by detailed pseudocode, outlines the Large Label Last (LLL) optimization strategy using a queue. This implementation maintains queue statistics (node count and distance sum) and implements a mechanism to reposition nodes whose distances exceed the queue's average to the rear, thereby optimizing the processing order.

## C    LLM AS JUDGE

The figure 8 illustrates a structured judgment prompt designed for Large Language Models (LLMs) serving as automated evaluators in ICC tasks. The prompt establishes a systematic framework for binary assessment of code implementations, emphasizing two primary evaluation criteria: instruction adherence and ground truth alignment. The evaluation protocol is formalized through a structured output format ([JUDGMENT][/JUDGMENT] and [REASON][/REASON] tags), enabling consistent and interpretable assessments. This prompt architecture specifically guides LLMs to focus on critical implementation aspects, including function definitions, data structures, algorithm steps, and control flow patterns, while maintaining a clear binary decision mechanism for determining implementation correctness. Such a structured approach facilitates reliable automated evaluation in code completion tasks, where precise assessment of implementation fidelity is crucial.

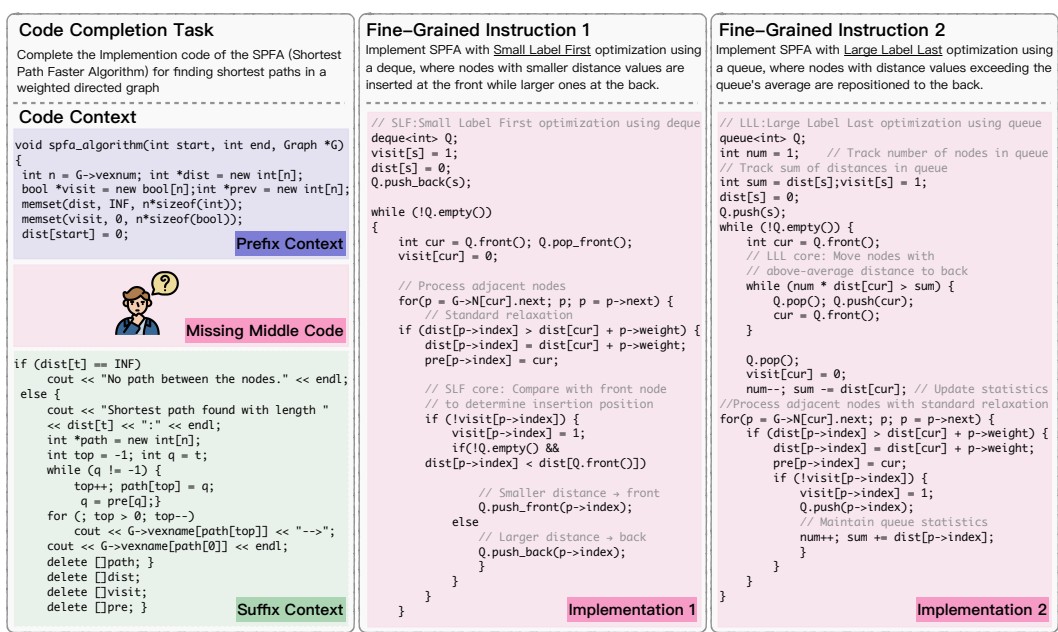

Figure 7: Example of Controllable Code Completion task requiring SPFA implementation with different optimization strategies (Small Label First vs. Large Label Last) based on distinct fine-grained instructions.

**Judgment Prompt**

As a code evaluator, assess whether the provided implementation follows the instruction and matches the implementation approach of the ground truth.

Focus on:

1) Instruction adherence: Does the implementation use the specified method/approach and contain all necessary components? This includes:

- Required function/class definitions

- Necessary data structures

- Key algorithm steps

- Essential control flow structures

- Critical variables and parameters

2) Ground truth alignment: Does it follow similar implementation strategy and logic flow as the ground truth solution?

Provide your evaluation in the following format:

[JUDGMENT]yes/no[/JUDGMENT]

[REASON]Brief explanation of your judgment (1-2 sentences)[/REASON]

Where:

- "yes": Implementation follows the instruction and matches the core approach of the ground truth

- "no": Implementation uses fundamentally different methods or structures from what was required

Figure 8: The illustration of judgment system prompt

```
implementation generation Prompt
You are a code helper. Your task is to create VERY DIFFERENT but FUNCTIONALLY IDENTICAL versions of a given code piece.

Two Key Requirements:
DIFFERENT: Each version must use a significantly different approach
IDENTICAL: All versions must work EXACTLY like the original code
Basic Rules:
* Put each version in [IMPi][/IMPi] tags
* Make at least 3 versions
* Must pass the same test cases
* Must handle the same edge cases
* Must have same input/output behavior
For Each Version:
* Use a unique implementation approach
* Maintain 100% functional equivalence
* Keep same error handling
* Keep same performance guarantees
Quality Check:
* Different: Clear differences in implementation style
* Same: All functional aspects must be identical
* Test: Should work the same in all situations
* Verify: Double-check all edge cases work
```

Figure 9: The illustration of implementation generation system prompt

# D CODE COMPLETION PERFORMANCE ON CONVENTIONAL BENCHMARKS

## D.1 PERFORMANCE ON CROSSCODEEVAL

The experimental results on CrossCodeEval showed in Table 3 demonstrate several noteworthy patterns across open-source and closed-source models. Among open-source models, we observe a general correlation between model size and performance, with larger models typically achieving better results. Notably, Qwen2.5-Coder-32B achieves state-of-the-art performance with an average EM score of 57.1The performance comparison between open-source and closed-source models reveals an interesting trend. Despite the extensive resources behind closed-source models, top-performing open-source models like Qwen2.5-Coder series demonstrate competitive or superior performance. For instance, Qwen2.5-Coder-32B outperforms all tested closed-source models, including GPT-4, Claude, and Gemini, across most metrics. This empirical evidence suggests that recent advances in open-source language models have achieved performance parity with, or even exceeded, their closed-source counterparts in code completion tasks.

Table 3: Performance of different approaches on the CrossCodeEval Tasks.

| Size | Model | Python | | Java | | TypeScript | | C# | | Average | |
|---|---|---|---|---|---|---|---|---|---|---|---|
| | | EM | ES | EM | ES | EM | ES | EM | ES | EM | ES |
| Open-Source Models | Qwen2.5-Coder-0.5B | 22.7 | 66.2 | 21.7 | 66.8 | 21.9 | 67.2 | 32.1 | 75.4 | 24.6 | 68.9 |
| | DS-Coder-1.3B-Base | 33.4 | 72.6 | 34.9 | 74.5 | 36.7 | 76.4 | 46.6 | 83.5 | 37.9 | 76.8 |
| | Qwen2.5-Coder-1.5B | 35.5 | 74.3 | 37.9 | 76.5 | 37.6 | 77.4 | 49.8 | 84.5 | 40.2 | 78.2 |
| | StarCoder2-3B | 11.0 | 62.7 | 11.6 | 69.7 | 8.8 | 75.8 | 8.2 | 71.2 | 9.9 | 69.8 |
| | Qwen2.5-Coder-3B | 38.4 | 76.1 | 42.8 | 79.8 | 41.6 | 80.5 | 56.7 | 87.1 | 44.9 | 80.9 |
| | StarCoder2-7B | 10.9 | 63.1 | 8.3 | 71.0 | 6.7 | 76.8 | 7.3 | 72.1 | 8.3 | 70.8 |
| | DS-Coder-6.7B-Base | 41.1 | 79.2 | 39.9 | 80.1 | 46.3 | 82.4 | 55.0 | 86.9 | 45.6 | 82.1 |
| | DS-Coder-V2-Lite-Base | 41.8 | 78.3 | 46.1 | 81.2 | 44.6 | 81.4 | 58.7 | 87.9 | 47.8 | 82.2 |
| | CodeQwen1.5-7B | 40.7 | 77.8 | 47.0 | 81.6 | 45.8 | 82.2 | 59.7 | 87.6 | 48.3 | 82.3 |
| | Qwen2.5-Coder-7B | 42.4 | 78.6 | 48.1 | 82.6 | 46.8 | 83.4 | 59.7 | 87.9 | 49.3 | 83.1 |
| | StarCoder2-15B | 28.2 | 70.5 | 26.7 | 71.0 | 24.7 | 76.3 | 25.2 | 74.2 | 26.2 | 73.0 |
| | Qwen2.5-Coder-14B | 47.7 | 81.7 | 54.7 | 85.7 | 52.9 | 86.0 | 66.4 | 91.1 | 55.4 | 86.1 |
| | CodeStral-22B | 49.3 | 82.7 | 44.1 | 71.1 | 51.0 | 85.0 | 53.7 | 83.6 | 49.5 | 80.6 |
| | DS-Coder-33B-Base | 44.2 | 80.4 | 46.5 | 82.7 | 49.2 | 84.0 | 55.2 | 87.8 | 48.8 | 83.7 |
| | Qwen2.5-Coder-32B | 49.2 | 82.1 | 56.4 | 86.6 | 54.9 | 87.0 | 68.0 | 91.6 | 57.1 | 86.8 |
| | DeepSeek-V3 | 37.1 | 69.9 | 42.8 | 71.5 | 33.2 | 66.9 | 42.8 | 72.7 | 39.0 | 70.2 |
| | DeepSeek-V3-0324 | 41.4 | 77.2 | 48.9 | 80.5 | 38.8 | 77.5 | 48.6 | 84.5 | 44.4 | 79.9 |
| | **Qwen2.5-Coder-32B-C[3]** | 47.4 | 81.1 | 56.5 | 86.6 | 54.2 | 86.4 | 65.5 | 90.8 | 55.9 | 86.2 |
| Closed-APIs | GPT-4o-2024-08-06 | 34.3 | 73.1 | 43.1 | 78.4 | 36.8 | 76.3 | 46.7 | 81.0 | 40.2 | 77.2 |
| | GPT-4o-2024-11-20 | 29.4 | 68.8 | 37.3 | 74.7 | 32.0 | 73.0 | 38.2 | 73.7 | 34.2 | 72.5 |
| | o1-2024-12-17 | 14.9 | 67.0 | 33.6 | 77.3 | 30.6 | 76.7 | 28.6 | 80.6 | 26.9 | 75.4 |
| | Claude3.5-Sonnet-20241022 | 45.2 | 79.6 | 49.3 | 84.3 | 42.8 | 81.2 | 52.5 | 84.1 | 47.5 | 82.3 |
| | Gemini-2.0-Flash | 38.7 | 69.0 | 48.2 | 77.9 | 41.5 | 76.9 | 47.0 | 79.0 | 43.8 | 75.7 |

## D.2 PERFORMANCE ON REPOEVAL

On the RepoEval benchmark (Table 4), Qwen2.5-Coder-32B achieves state-of-the-art performance among all tested models, both open-source and closed-source, with an average EM score of 51.6% and ES score of 78.5%. Qwen2.5-Coder-32B-C[3] maintains comparable performance with an average EM of 51.8% and ES of 77.0%, demonstrating clear advantages over leading closed-source models like Claude3.5-Sonnet and GPT-4o

## D.3 PERFORMANCE ON CROSSCODELONGEVAL

On the CrossCodeLongEval benchmark (Table 5), Qwen2.5-Coder-32B achieves the best overall performance among all models, with an average EM score of 36.9% and ES score of 66.4%. This performance slightly exceeds that of leading closed-source models, including Claude3.5-Sonnet (EM: 32.4%, ES: 63.2%) and other commercial APIs.

Table 4: Performance of different approaches on the RepoEval Tasks.

| Size | Model | Line | | Function | | API | | Average | |
|---|---|---|---|---|---|---|---|---|---|
| | | EM | ES | EM | ES | EM | ES | EM | ES |
| Open-Source Models | Qwen2.5-Coder-0.5B | 44.2 | 72.6 | 4.6 | 48.0 | 35.6 | 68.5 | 28.1 | 63.0 |
| | DS-Coder-1.3B-Base | 58.7 | 80.4 | 6.2 | 48.8 | 45.8 | 75.0 | 36.9 | 68.1 |
| | Qwen2.5-Coder-1.5B | 59.8 | 82.6 | 10.6 | 52.4 | 51.0 | 80.1 | 40.5 | 71.7 |
| | StarCoder2-3B | 22.3 | 67.4 | 3.1 | 51.6 | 20.6 | 70.1 | 15.3 | 63.0 |
| | Qwen2.5-Coder-3B | 64.9 | 85.0 | 12.3 | 55.8 | 54.7 | 81.3 | 44.0 | 74.0 |
| | StarCoder2-7B | 19.5 | 67.6 | 4.0 | 53.5 | 19.1 | 72.8 | 14.2 | 64.7 |
| | DS-Coder-6.7B-Base | 63.1 | 85.5 | 9.9 | 53.3 | 52.3 | 81.7 | 41.7 | 73.5 |
| | DS-Coder-V2-Lite-Base | 66.5 | 85.4 | 10.8 | 53.9 | 53.1 | 81.3 | 43.4 | 73.5 |
| | CodeQwen1.5-7B | 59.7 | 81.5 | 4.8 | 44.3 | 46.1 | 77.5 | 36.9 | 67.8 |
| | Qwen2.5-Coder-7B | 67.3 | 86.1 | 13.2 | 55.2 | 58.4 | 83.9 | 46.3 | 75.1 |
| | StarCoder2-15B | 30.9 | 62.5 | 5.5 | 43.7 | 21.7 | 60.3 | 19.4 | 55.5 |
| | Qwen2.5-Coder-14B | 74.3 | 90.1 | 14.1 | 59.5 | 63.4 | 87.3 | 50.6 | 79.0 |
| | CodeStral-22B | 40.9 | 51.7 | 9.9 | 49.2 | 24.8 | 40.8 | 30.0 | 46.6 |
| | DS-Coder-33B-Base | 66.5 | 86.6 | 10.3 | 52.9 | 54.2 | 83.5 | 43.7 | 74.3 |
| | Qwen2.5-Coder-32B | 76.1 | 90.5 | 13.6 | 57.5 | 65.1 | 87.6 | 51.6 | 78.5 |
| | DeepSeek-V3 | 47.2 | 63.1 | 18.5 | 49.3 | 47.6 | 68.9 | 37.7 | 60.4 |
| | DeepSeek-V3-0324 | 60.4 | 77.5 | 19.6 | 49.2 | 57.5 | 78.0 | 45.8 | 68.2 |
| | **Qwen2.5-Coder-32B-C$^3$** | 74.8 | 90.2 | 13.0 | 52.4 | 67.7 | 88.3 | 51.8 | 77.0 |
| Closed-APIs | GPT-4o-2024-08-06 | 50.7 | 69.1 | 13.6 | 42.9 | 47.3 | 72.6 | 37.2 | 61.5 |
| | GPT-4o-2024-11-20 | 37.5 | 57.0 | 5.1 | 38.5 | 34.6 | 60.8 | 25.7 | 52.1 |
| | o1-2024-12-17 | 57.5 | 71.9 | 20.2 | 55.8 | 55.8 | 77.4 | 44.5 | 68.4 |
| | Claude3.5-Sonnet-20241022 | 61.9 | 80.1 | 22.0 | 55.1 | 60.0 | 81.1 | 48.0 | 72.1 |
| | Gemini-2.0-Flash | 59.0 | 74.5 | 16.0 | 46.7 | 58.1 | 80.4 | 44.4 | 67.2 |

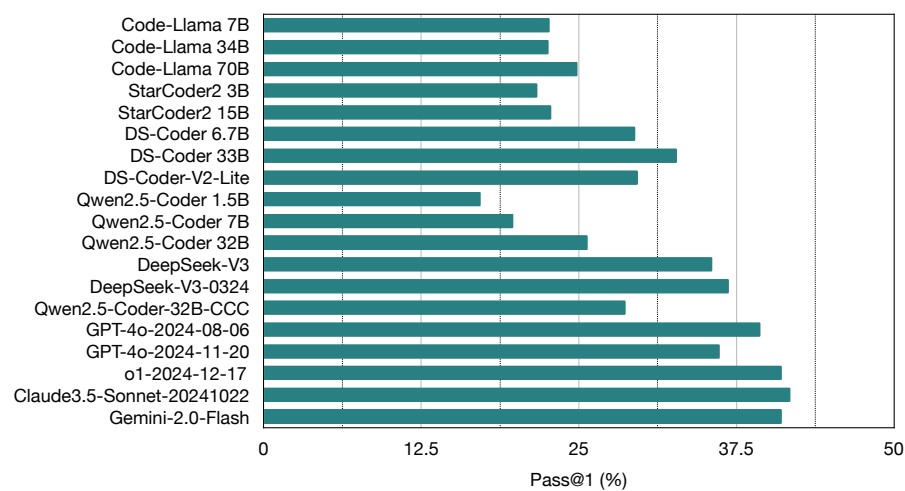

Figure 10: Performance of different approaches on the ExecRepoBench Tasks.

## D.4 PERFORMANCE ON SAFIM

On the SAFIM benchmark (Table 6), Qwen2.5-Coder-32B achieves the highest average pass rate of 71.2% across all evaluated models. The model demonstrates strong performance across all three categories: Algorithm (61.1%), Control (74.6%), and API (77.7%). Its C³-tuned variant maintains competitive performance with an average pass rate of 69.5%, significantly outperforming closed-source models like Gemini-2.0-Flash (64.4%) and Claude3.5-Sonnet (63.6%).

Table 5: Performance of different approaches on the CrossCodeLongEval Tasks.

| Size | Model | Chunk Completion | | Function completion | | Average | |
|---|---|---|---|---|---|---|---|
| | | *EM* | *ES* | *EM* | *ES* | *EM* | *ES* |
| Open-Source Models | Qwen2.5-Coder-0.5B | 29.8 | 64.2 | 9.5 | 38.0 | 19.7 | 51.1 |
| | DS-Coder-1.3B-Base | 40.6 | 71.9 | 9.6 | 39.4 | 25.1 | 55.7 |
| | Qwen2.5-Coder-1.5B | 44.2 | 73.9 | 12.4 | 44.4 | 28.3 | 59.2 |
| | StarCoder2-3B | 18.5 | 62.0 | 10.2 | 39.2 | 14.3 | 50.6 |
| | Qwen2.5-Coder-3B | 46.6 | 76.1 | 13.5 | 46.4 | 30.0 | 61.3 |
| | StarCoder2-7B | 19.4 | 63.6 | 10.2 | 40.0 | 14.8 | 51.8 |
| | DS-Coder-6.7B-Base | 48.4 | 78.2 | 10.7 | 42.4 | 29.6 | 60.3 |
| | DS-Coder-V2-Lite-Base | 49.5 | 77.1 | 11.4 | 43.1 | 30.4 | 60.1 |
| | CodeQwen1.5-7B | 48.2 | 77.5 | 6.4 | 30.6 | 27.3 | 54.1 |
| | Qwen2.5-Coder-7B | 52.4 | 79.3 | 14.4 | 48.4 | 33.4 | 63.8 |
| | StarCoder2-15B | 21.3 | 53.7 | 7.8 | 30.5 | 14.6 | 42.1 |
| | Qwen2.5-Coder-14B | 56.9 | 81.8 | 15.4 | 49.8 | 36.1 | 65.8 |
| | CodeStral-22B | 56.7 | 81.8 | 10.5 | 37.8 | 33.6 | 59.8 |
| | DS-Coder-33B-Base | 52.0 | 79.9 | 11.9 | 44.3 | 32.0 | 62.1 |
| | Qwen2.5-Coder-32B | 57.3 | 82.1 | 16.4 | 50.8 | 36.9 | 66.4 |
| | DeepSeek-V3 | 35.1 | 57.3 | 15.7 | 49.8 | 25.4 | 53.5 |
| | DeepSeek-V3-0324 | 44.8 | 69.4 | 16.9 | 50.9 | 30.9 | 60.2 |
| | **Qwen2.5-Coder-32B-C$^3$** | 47.6 | 69.1 | 10.5 | 52.0 | 29.1 | 60.5 |
| Closed-APIs | GPT-4o-2024-08-06 | 44.8 | 71.2 | 15.3 | 53.3 | 30.1 | 62.2 |
| | GPT-4o-2024-11-20 | 41.9 | 67.9 | 10.8 | 48.4 | 26.4 | 58.2 |
| | o1-2024-12-17 | 39.9 | 62.7 | 13.3 | 50.5 | 26.6 | 56.6 |
| | Claude3.5-Sonnet-20241022 | 47.2 | 72.7 | 17.5 | 53.7 | 32.4 | 63.2 |
| | Gemini-2.0-Flash | 42.4 | 65.6 | 15.6 | 48.0 | 29.0 | 56.8 |

# E  ADDITIONAL EXPERIMENTAL ANALYSIS

## E.1  MODEL PREFERENCE ANALYSIS

In this section, we analyze the generation preferences of different models in code completion tasks. By calculating token counts for both completed middle code and additional explanations (Commentary) on $C^3$-Bench, as shown in Figure 11, we observe distinct patterns among models. Claude Series and DeepSeek Series models tend to generate more commentary beyond code completion, while GPT Series, o1-Series, and models like Qwen2.5-Coder-14B-Instruct and Qwen2.5-Coder-3B-Instruct focus solely on completion without additional commentary.

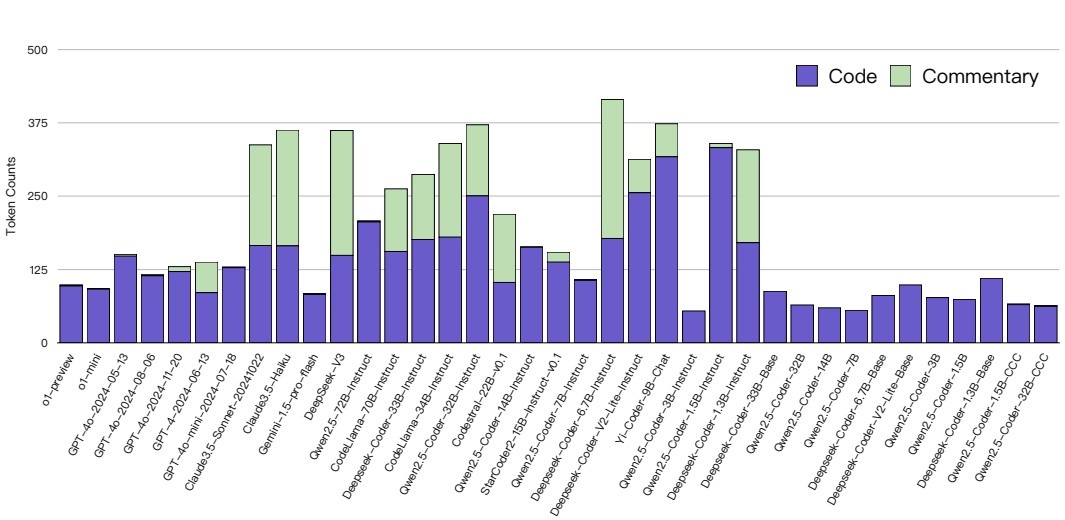

Figure 11: Token Counts of different model generations on $C^3$-Bench. Code represents the token count of middle code completions, Commentary represents the token count of additional explanations and descriptions provided by models.

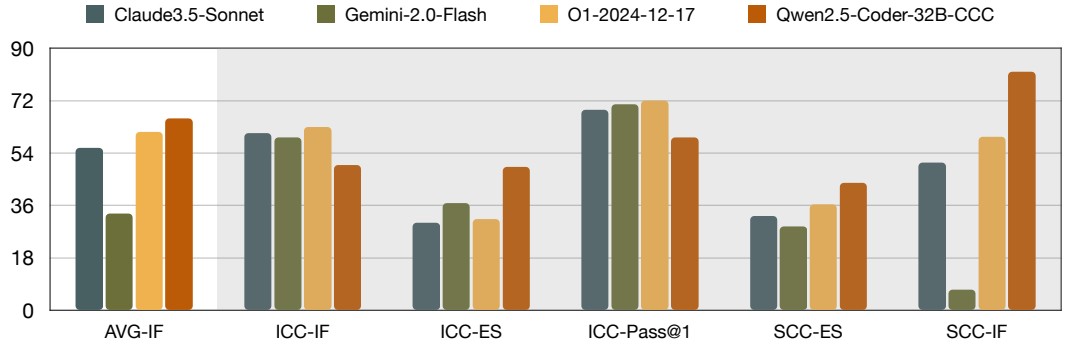

Figure 12: Comparison between model performance on ICC and SCC tasks.

Table 6: Performance of different approaches on the SAFIM Tasks.

| Size | Model | SAFIM | | | |
|---|---|---|---|---|---|
| | | *Algo.* | *Control* | *API* | *Average* |
| Open-Source Models | Qwen2.5-Coder-0.5B | 24.3 | 37.9 | 49.7 | 37.3 |
| | DS-Coder-1.3B-Base | 39.3 | 52.6 | 62.6 | 51.5 |
| | Qwen2.5-Coder-1.5B | 37.3 | 39.6 | 66.5 | 47.8 |
| | StarCoder2-3B | 19.9 | 29.1 | 67.4 | 38.8 |
| | Qwen2.5-Coder-3B | 45.7 | 59.0 | 68.1 | 57.6 |
| | StarCoder2-7B | 38.5 | 38.7 | 70.6 | 49.3 |
| | DS-Coder-6.7B-Base | 52.8 | 64.9 | 71.6 | 63.1 |
| | DS-Coder-V2-Lite-Base | 56.3 | 69.9 | 75.5 | 67.2 |
| | CodeQwen1.5-7B | 37.3 | 58.3 | 71.9 | 55.8 |
| | Qwen2.5-Coder-7B | 50.5 | 58.1 | 73.9 | 60.8 |
| | StarCoder2-15B | 36.9 | 55.9 | 70.3 | 54.4 |
| | Qwen2.5-Coder-14B | 57.1 | 70.8 | 75.8 | 67.9 |
| | DS-Coder-33B-Base | 59.1 | 69.8 | 74.2 | 67.7 |
| | Qwen2.5-Coder-32B | 61.1 | 74.6 | 77.7 | 71.2 |
| | DeepSeek-V3 | 60.5 | 55.8 | 64.1 | 60.1 |
| | DeepSeek-V3-0324 | 53.3 | 68.1 | 65.3 | 62.2 |
| | **Qwen2.5-Coder-32B-C**[3] | 60.9 | 73.4 | 68.3 | 67.5 |
| Closed-APIs | GPT-4o-2024-08-06 | 47.9 | 64.2 | 54.9 | 55.7 |
| | GPT-4o-2024-11-20 | 59.5 | 65.2 | 58.6 | 61.1 |
| | o1-2024-12-17 | 62.6 | 67.1 | 65.9 | 65.2 |
| | Claude3.5-Sonnet-20241022 | 60.6 | 61.3 | 68.9 | 63.6 |
| | Gemini-2.0-Flash | 62.2 | 66.8 | 64.1 | 64.4 |

```
               Implementation Control Completion Prefix Code
                    def DFS(start):
                        nodes=set()
                        stack=[start]
                        while stack:
                            parent=stack.pop()
                            if(not visited[parent]):
                                nodes.add(parent)
                                visited[parent]=True
                                for child in graph[parent]:
                                    if (not visited[child]):
                                        stack.append(child)
                                    else:
                                        if child not in nodes and child!=s:
                                            return child
                            else:
                                if parent not in nodes and parent != s:
                                    return parent
                        return -1
```

Figure 13: Prefix Code of the ICC task example

## F  C³-BENCH EXAMPLES

### F.1  IMPLEMENTATION CONTROL COMPLETION EXAMPLE

In this section, we introduce an example ICC task from C³-Bench. This example focuses on finding two different paths in a labyrinth from a start node to an end node, where paths can only share the start and end points. The task requires inputs of n vertices, m edges, and a starting point s, and outputs either "Possible" with two valid paths or "Impossible".

The task structure consists of multiple components. The prefix code, illustrated in Figure 13, contains a helper function for initial DFS exploration to identify potential end points. The suffix code, shown in Figure 14, manages input processing, result validation, and output formatting. The middle implementation can be achieved through three distinct approaches: an iterative DFS using a stack (Figure 15), a recursive DFS with parent pointers (Figure 16), and a BFS implementation using a queue (Figure 17). These implementations, while functionally equivalent, demonstrate different approaches to path finding and parent tracking. The iterative DFS maintains explicit stack control, the recursive DFS offers cleaner code structure, and the BFS provides shortest path guarantees, each with its own trade-offs in terms of memory usage and code clarity.

### F.2  SCALE CONTROL COMPLETION EXAMPLE

We present an example of a Scale-Control Completion (SCC) task from C³-Bench. As shown in Figure 18 and Figure 19. The task specifically requires generating only a single for statement block, with no additional code allowed. Figure 20 shows the system instruction and the implementation that strictly adheres to this scope constraint, demonstrating precise control over code generation granularity.

```
┌─ Implementation Control Completion Suffix Code ──────────┐
│                                                           │
│   def get_path(node):            for child in graph[s]:   │
│       path=[]                        end=DFS(child)       │
│       while node!=-1:                if end!=-1:          │
│           path.append(node)              visited = [False] * n │
│           node=parent_list[node]         parent_list=[-1]*n │
│       path.reverse()                     visited[s]=True  │
│       return path                        ans=[]           │
│   n,m,s=map(int,input().split())         for child in graph[s]: │
│   s-=1                                       if DFS_get_path(child): │
│   graph=[[] for _ in range(n)]                   ans.append([s]+get_path(end)) │
│   for _ in range(m):                     if len(ans)==2: │
│       a,b=map(int,input().split())           break        │
│       a-=1                           print("Possible")    │
│       b-=1                           for i in ans:        │
│       graph[a].append(b)                 print(len(i))    │
│   visited=[False]*n                      print(*[j+1 for j in i]) │
│   visited[s]=True                    break                │
│                                  else:                    │
│                                      print("Impossible")  │
│                                                           │
└───────────────────────────────────────────────────────────┘
```

Figure 14: Suffix Code of the ICC task example

```
┌─ Implementation 1 ───────────────────────────────────────┐
│                                                           │
│  Instruction：                                            │
│   Use iterative DFS with a stack to find path from start to end node │
│                                                           │
│  Middle Code：                                            │
│                                                           │
│               def DFS_get_path(start):                    │
│                   stack=[start]                           │
│                   parent_list[start]=-1                   │
│                   while stack:                            │
│                       parent=stack.pop()                  │
│                       if parent==end:                     │
│                           visited[end]=False              │
│                           return True                     │
│                       if(not visited[parent]):            │
│                           visited[parent]=True            │
│                           for child in graph[parent]:     │
│                               if (not visited[child]):    │
│                                   stack.append(child)     │
│                                   parent_list[child]=parent │
│                   return False                            │
│                                                           │
└───────────────────────────────────────────────────────────┘
```

Figure 15: Instruction and Implementation code 1 of the ICC task example

**——————Implementation 2——————**

**Instruction：**

Use recursive DFS to find paths from start to end node,
maintaining parent pointers for path reconstruction

**Middle Code：**

```
def DFS_get_path(start):
    if start == end:
        visited[end] = False
        return True

    visited[start] = True
    for child in graph[start]:
        if not visited[child]:
            parent_list[child] = start
            if DFS_get_path(child):
                return True
    return False
```

Figure 16: Instruction and Implementation code 2 of the ICC task example

**——————Implementation 3——————**

**Instruction：**

Use BFS with a queue to find shortest paths from start to end node,
storing parent pointers for path reconstruction

**Middle Code：**

```
def BFS_get_path(start):
    queue = [(start, -1)]
    while queue:
        curr, prev = queue.pop(0)
        if curr == end:
            visited[end] = False
            return True

        if not visited[curr]:
            visited[curr] = True
            parent_list[curr] = prev
            for child in graph[curr]:
                if not visited[child]:
                    queue.append((child, curr))
    return False
```

Figure 17: Instruction and Implementation code 3 of the ICC task example

─── Scale Control Completion **Prefix Code** ───

```
clean_data = data.dropna()

# Feature and target selection
# Assuming 'EnergyConsumption' is the target variable and others are features
target_variable = 'EnergyConsumption'
features = clean_data.columns.drop(target_variable)

# Feature selection using Scikit-learn
# Selecting the top 3 features that have the highest correlation with the target variable
k_best_features = 3
selector = SelectKBest(score_func=f_regression, k=k_best_features)
selected_features = selector.fit_transform(clean_data[features], clean_data[target_variable])
selected_feature_names = clean_data[features].columns[selector.get_support()]

print("\nSelected features:")
print(selected_feature_names)

# Splitting the data into training and testing sets
# Using TimeSeriesSplit for cross-validation
n_splits = 3
tscv = TimeSeriesSplit(n_splits=n_splits)

for train_index, test_index in tscv.split(selected_features):
```

Figure 18: Prefix Code of the SCC task example

─── Scale Control Completion **Suffix Code** ───

```
        # Fitting a Vector Autoregression (VAR) model
        model = sm.tsa.VAR(clean_data)
        results = model.fit(maxlags=5, ic='aic')

        # Displaying the summary of the VAR model results
        print("\nVAR Model Results:")
        print(results.summary())
```

Figure 19: Suffix Code of the SCC task example

─────── Implementation ───────

**Instruction：**

Just Complete the for statement block in the prefix code.

**Middle Code：**

```
X_train, X_test = selected_features[train_index], selected_features[test_index]
y_train, y_test = clean_data[target_variable].values[train_index], clean_data[target_variable].values[test_index]

# Fitting a linear regression model using Scikit-learn
lr_model = LinearRegression()
lr_model.fit(X_train, y_train)

# Predicting the target variable for the test set
y_pred = lr_model.predict(X_test)

# Calculating the Mean Squared Error (MSE) for the model
mse = mean_squared_error(y_test, y_pred)
print(f"\nMean Squared Error for split {tscv.split}: {mse}")
```

Figure 20: Instruction and Implementation code of the SCC task example

