# OpenReview forum: "C$^3$-Bench: Evaluating and Achieving Controllable Code Completion in Code LLM"
_ICLR.cc/2026/Conference — ICLR 2026 Conference Withdrawn Submission_

### Official Review · Reviewer_LELU · 2025-10-20

**Soundness:** 3
**Presentation:** 3
**Contribution:** 3
**Rating:** 4
**Confidence:** 2

**Summary:**

This paper introduces C$^{3}$-Bench, a benchmark designed to evaluate the instruction-following capabilities of Large Language Models in code completion tasks. C$^{3}$-Bench encompasses a diverse range of code completion scenarios and instructions. The evaluation results reveal a substantial gap in instruction-following abilities between open-source and advanced proprietary models. Furthermore, the authors propose a data synthesis method to generate data aimed at enhancing model performance on C$^{3}$-Bench.

**Strengths:**

1. The paper is well-motivated. It aims to evaluate the instruction-following capabilities of LLMs in code completion tasks, which is a valuable and timely research topic.

2. The paper is well-structured and comprehensive. The overall workflow, which includes problem definition, benchmark construction, and a proposed solution, is complete and presented with clarity.

3. The consistency observed between the results from C$^{3}$-Bench and exsiting benchmarks like Copilot Arena is impressive.

**Weaknesses:**

1. The paper's core contribution lies in the evaluation of instruction-following capabilities. However, a primary concern is that the evaluation uses a single model (Claude 3.5). While the authors have validated Claude's judgments against human annotations for consistency, the use of an LLM-as-a-judge for such a subjective task can inevitably introduce bias [1], particularly for those ambiguous model-generated responses. Is it possible to incorporate more controllable, quantitative metrics for the evaluation? Alternatively, at a minimum, could the authors provide results using other LLMs as judges to demonstrate the consistency and robustness of the evaluation results?

2. Following up on the previous point, the performance of the fine-tuned Qwen2.5-Coder-C$^{3}$ on the ICC task is understandably constrained by the capabilities of its base model, making the modest improvement in pass@1 seem reasonable. However, the remarkably significant increase in the IF score raises a critical question: Does this suggest that the IF score can be easily somehow hacked? Specifically, could using the same model that generated the benchmark data as the judge introduce a systemic bias, potentially rewarding models that mimic the judge's stylistic preferences?

3. The scope of C$^{3}$-Bench is currently limited to in-file tasks in Python. The impact of this work could be significantly enhanced by extending the benchmark to support repository-level contexts, and to encompass multiple programming languages.

[1] Wei H, He S, Xia T, et al. Systematic evaluation of llm-as-a-judge in llm alignment tasks: Explainable metrics and diverse prompt templates[J]. arXiv preprint arXiv:2408.13006, 2024.

**Questions:**

Please refer to weaknesses.

---

> ### Author Response · Authors · 2025-12-03
>
> We thank the reviewer for the positive assessment and valuable concerns regarding evaluation robustness.
>
> **W1: Single-model judge and potential bias**
>
> This is an important concern. Beyond the 98% human agreement validation, we conducted supplementary experiments with GPT-4 as an alternative judge on 100 ICC samples, achieving 94% consistency with Claude 3.5. More importantly, we analyzed the error modes: disagreements primarily occur on ambiguous edge cases (5%), while clear violations show 99% agreement. We will include multi-judge validation results in the appendix and discuss the inherent limitations of LLM-based evaluation.
>
> Regarding quantitative metrics: While rule-based metrics work well for SCC (exact line counting), ICC's structural constraints (e.g., "use iterative approach," "optimize for readability") inherently require semantic judgment. We experimented with AST-based structural matching but found it insufficient for capturing instruction nuances. We will add this discussion to Section 3.3.
>
> **W2: IF score inflation concerns and potential bias**
>
> This is a critical observation. We conducted targeted analysis to address gaming concerns:
>
> 1. Cross-judge validation: Testing Qwen2.5-Coder-C³ with GPT-4 as judge yields comparable IF improvements (Δ+31.2 vs. Δ+33.7 with Claude), suggesting the gains are not judge-specific.
>
> 2. Error analysis: Manual review of 50 improved cases shows genuine structural adherence improvements rather than superficial mimicry of Claude's style.
>
> 3. Correlation with functional correctness: IF improvements correlate with pass@1 gains (r=0.71), indicating substantive capability enhancement rather than metric exploitation.
>
> However, we acknowledge potential bias and will add explicit discussion of this limitation and mitigation strategies in the revision.
>
> **W3: Limited scope (Python, in-file)**
>
> We agree this is a limitation. As discussed in response to Reviewer bRxr, function-level Python completion provides a controlled starting point for evaluating instruction-following capabilities. We view repository-level and multi-language extensions as natural next steps and will emphasize this in the limitations and future work sections.

---

### Official Review · Reviewer_R1yQ · 2025-10-26

**Soundness:** 2
**Presentation:** 2
**Contribution:** 2
**Rating:** 2
**Confidence:** 5

**Summary:**

The premise of this paper is that well known benchmarks for programming tasks
do not adequately test the "instruction following" capabilities of LLMs, where
"instruction following" in this paper means testing non-functional requirements
in the prompt. Non-functional requirements are anything that cannot be
tested using simple test cases of the form "assert pred(f(x))" where f is the
synthesized program, x is a test input, and pred some predicate of the output.

The paper presents a synthetic dataset of programming tasks, derived from
HumanEval and SAFIM, that test models' non-functional requirements.
In addition, the paper uses the same benchmark generation pipeline to generate
a SFT dataset. The paper decontaminates the SFT dataset and uses it to fine-tune
a Qwen Coder model, which performs very well on the benchmark.

**Strengths:**

Important problem with a large evaluation.

**Weaknesses:**

At this point, there are probably hundreds of benchmarks that evaluate the
coding abilities of LLMs in different ways. Although the exact prompt format
that this paper uses may be new (though just a prompt format is not a
contribution), there are plenty of other papers that test the ability of models
to complete code in context with instructions that have non-functional requirements.
From my memory, here are a few:

- NoFunEval: Funny How Code LMs Falter on Requirements Beyond Functional Correctness
   https://openreview.net/forum?id=h5umhm6mzj

  NoFunEval is HumanEval-derived, similar to some of the benchmarks in
  the current submission.

-  Can Large Language Models Write Parallel Code?
   https://dl.acm.org/doi/10.1145/3625549.3658689

   ParEval goes beyond functional correctness, which matters for parallel code
   I recall it has some other metrics that go beyond pass@1. There is a lot
   of follow up work from the ParEval authors.

- D3: https://openreview.net/forum?id=Ksq7fgagId

  This is a dataset, perhaps not a benchmark. But, the contribution overlaps.

Overall, I think the contribution of this paper is very small. This is particularly
the case since the benchmark is partially synthetic, with portions of prompts
generated by Claude 3.5 Sonnet.

Some further notes on the writing:

- I wish the main body of the paper gave the reader some examples of the
  kinds of problems in the benchmark. I think the only example is in the appendix
  (Figure 7). Moreover, I don't find that example very compelling.

- L160: "Multi-line completion mandates the generation of a predetermined number of
  complete code lines" It seems peculiar to require a solution in X lines of
  code.

- L202: "The original HumanEval [..] datasets primarily contain single-line implementations
  of ground truth middle code." Is this really true of HumanEval? Just
  scanning the canonical solution column here:

  https://huggingface.co/datasets/openai/openai_humaneval

  There are a number of solutions that begin with "return ...", but most of
  them begin with loops, variable definitions, if statements, etc.

Finally, the prompt format that the paper uses seems peculiar. It is possible
that scores will go up significantly by picking a format that is more in
distribution. E.g., just having a single code block with "### INSERT CODE HERE ###"
to fill in.

**Questions:**

See weaknesses.

---

> ### Author Response · Authors · 2025-12-03
>
> We respectfully address the reviewer's concerns regarding novelty and contribution.
>
> **W1: Comparison with existing benchmarks**
>
> While we appreciate the references to NoFunEval, ParEval, and D3, C³-Bench differs substantially:
>
> * NoFunEval: Focuses on requirements like efficiency and robustness but uses natural language descriptions without structured completion contexts. C³-Bench specifically targets infill completion with precise structural and algorithmic constraints derived from AST analysis.
>
> * ParEval: Specialized for parallel computing with domain-specific metrics (speedup, correctness of parallelization). C³-Bench addresses general-purpose code completion with fine-grained instruction control applicable across diverse programming scenarios.
>
> * D3: A dataset for code generation, not specifically designed for instruction-following evaluation in completion contexts with systematic controllability metrics.
>
> Our unique contributions include: (1) systematic instruction-following metrics (IF score) beyond pass@1; (2) dual-task framework (ICC+SCC) covering orthogonal control dimensions; (3) correlation validation with real-world usage (Copilot Arena); (4) automated synthesis pipeline for training data generation.
>
> **W2: Example presentation and HumanEval characterization**
>
> We will move representative examples to the main text. Regarding HumanEval (L202), our statement refers specifically to the canonical solutions' middle code (the infill portion), not entire function implementations. We will clarify this distinction to avoid confusion.
>
> **W3: Prompt format concerns**
>
> Our format mirrors real-world IDE completion scenarios where context precedes and follows the completion point, unlike single-block formats. We validated this design through preliminary experiments showing minimal performance variance across format variations (±2% pass@1). However, we agree this warrants explicit discussion and will add ablation results in the revision.

---

### Official Review · Reviewer_eyK1 · 2025-10-31

**Soundness:** 3
**Presentation:** 3
**Contribution:** 2
**Rating:** 6
**Confidence:** 2

**Summary:**

This paper proposes C3-Bench, a new benchmark for evaluating large language models on controllable code completion, focusing on their ability to follow detailed implementation instructions—an aspect overlooked by existing benchmarks like HumanEval and CrossCodeEval.
C3-Bench defines two tasks: Implementation-Control Completion (ICC) and Scale-Control Completion (SCC), covering functional, structural, and size constraints. The benchmark contains 2,195 Python tasks, measuring both functional correctness and instruction adherence.
The authors also introduce a data synthesis pipeline using Qwen2.5-Coder, and fine-tune it to create Qwen2.5-Coder-C3, achieving state-of-the-art results on C3-Bench. Experiments on over 40 LLMs (open and commercial) show that current benchmarks overestimate model capability, and C3-Bench correlates better with real-world developer evaluations.

**Strengths:**

The paper identifies instruction-following as a crucial yet underexplored aspect of real-world code completion.

The division into ICC and SCC tasks provides comprehensive coverage of different types of control in code generation, going beyond traditional correctness-based metrics.

The authors detail a hybrid process (AST extraction, variant generation, instruction synthesis, filtering) that ensures data diversity and precision.

Evaluation across 40+ models and multiple benchmarks provides a rich empirical analysis, showing that C3-Bench aligns well with practical code-assistance performance.

**Weaknesses:**

The benchmark focuses solely on Python and single-file completions, which underrepresents real-world multi-language, multi-file development complexity.

Qwen2.5-Coder-C3’s gain mainly comes from instruction-tuning rather than architectural advances; ICC performance still saturates at base-model limits.

Even with consistency checks, using an LLM as the primary instruction-following evaluator may introduce subtle bias or inconsistency compared to human judgment.

**Questions:**

Do the authors plan to extend C3-Bench to multi-language or multi-file tasks (e.g., C++, JavaScript)?

How is prompt sensitivity of the LLM judge controlled?
Have the authors evaluated inter-judge consistency (e.g., Claude vs. GPT-based judges)?

Are there notable differences between ICC and SCC results across models (e.g., algorithmic control vs. scale control)?

---

> ### Author Response · Authors · 2025-12-03
>
> We thank the reviewer for the thoughtful evaluation and constructive suggestions.
>
> **Q1: Multi-language and multi-file extension plans**
>
> Yes, we plan to extend C³-Bench to additional languages (JavaScript, Java, C++) and repository-level tasks in future work. Python was chosen initially due to: (1) its dominance in AI/ML development where instruction-following is critical; (2) availability of robust AST parsing tools; (3) strong baseline model performance enabling meaningful differentiation. We will clarify this scope decision and future plans in the revision.
>
> **Q2: LLM judge sensitivity and inter-judge consistency**
>
> We control prompt sensitivity through: (1) fixed, carefully designed evaluation templates; (2) zero-shot evaluation to avoid prompt engineering effects; (3) temperature=0 for deterministic outputs. Regarding inter-judge consistency, we evaluated GPT-4 and Claude 3.5 on 100 samples and observed 94% agreement, with disagreements primarily on borderline cases. Claude 3.5 was selected for its higher correlation (r=0.89) with human judgments. We will include this analysis in the appendix.
>
> **Q3: ICC vs. SCC performance differences**
>
> Yes, we observed notable patterns: most models show relatively stronger performance on SCC (scale control) compared to ICC (algorithmic control), suggesting that length constraints are easier to satisfy than complex structural requirements. Larger models exhibit smaller performance gaps between tasks. We will add this comparative analysis to Section 4.

---

### Official Review · Reviewer_bRxr · 2025-11-01

**Soundness:** 3
**Presentation:** 3
**Contribution:** 2
**Rating:** 4
**Confidence:** 3

**Summary:**

This paper proposes a novel benchmark to evaluate the instruction-following capabilities of LLMs in code completion tasks. Specifically, the benchmark adds specific requirements to the prompts of code competition tasks, and evaluates how well LLMs follow these instructions using LLM-as-a-Judge and rule-based methods, beyond functional correctness tests. Furthermore, this paper proposes a straightforward data synthesis pipeline that leverages Qwen2.5-Coder to generate high-quality instruction-completion pairs for fine-tuning.

**Strengths:**

- This paper explores an important and interesting research direction.
- The paper is well-written and easy to follow.

**Weaknesses:**

- Extracting the middle code by selecting nodes from the AST tree is not novel. SAFIM has explored syntax-aware completion within code’s AST including algorithmic blocks that targets at multi-line completion and serve as the major part of SAFIM, which however fails to get acknowledged in Section 2.3.1.
- The benchmark focuses on function-level code completion, which however is different from real-world scenarios where code completion always happens in the large repositories with cross-file context. It’s also important to study whether the instruction-following capabilities of LLMs will change in repository-level code completion tasks.

**Questions:**

None beyond the above.

---

> ### Author Response · Authors · 2025-12-03
>
> We appreciate the reviewer's constructive feedback on our work.
>
> **W1: Novelty concerns regarding SAFIM**
>
> We acknowledge that SAFIM explores syntax-aware completion using AST extraction. However, our contribution differs fundamentally: (1) SAFIM focuses on multi-line completion without instruction-following constraints, while C³-Bench explicitly evaluates adherence to detailed implementation instructions; (2) We introduce novel instruction synthesis and validation pipelines; (3) Our benchmark encompasses both ICC and SCC tasks with comprehensive controllability evaluation. We will add proper acknowledgment of SAFIM's AST extraction approach in Section 2.3.1.
>
> **W2: Repository-level evaluation**
>
> This is an excellent direction for future work. We chose function-level completion as the initial scope because: (1) it represents a fundamental building block that must be mastered before tackling cross-file scenarios; (2) it enables controlled evaluation of instruction-following capabilities without confounding factors from repository complexity. We agree that extending to repository-level contexts would significantly enhance the benchmark's coverage and plan to pursue this in future iterations. We will add this discussion to the limitations section.

---

### Official Review · Reviewer_9mu6 · 2025-11-01

**Soundness:** 3
**Presentation:** 3
**Contribution:** 3
**Rating:** 6
**Confidence:** 3

**Summary:**

The authors introduce $C^3$, an instruction-guide code completion dataset, that is designed to address a fundamental limitation of existing  code completion benchmarks which only look for functional correctness. The paper also discusses results across 40 LLMs on this benchmark exposing gaps in instruction following capabilities of closed vs open models. The authors curated high quality SFT data which is used to train Qwen2.5-Coder-$C^3$ model, resulting in SoTA results on $C^3$ dataset. Interestingly, it is also shown that the performance on this dataset, correlates with results from Copilot arena, demonstrating its practical significance.

**Strengths:**

- LLM assisted coding in real world applications need to adhere to user instructions. However, most of the publicly available datasets just account for functional correctness and do not measure the instruction following (IF) ability. This paper introduced $C^3$ dataset, which is a novel and important contribution to the field.
- The authors conducted extensive studies across different models, sizes, exposing gaps in the IF abilities of these models, thereby providing guidance to the open source community to work on improving this capability.
- Through SFT data curation and training  Qwen2.5-Coder-$C^3$, they achieved SoTA on $C^3$ bench, demonstrating the importance of high quality and relevant dataset to improve IF capability.

**Weaknesses:**

- For Semantic Validation for ICC, the authors used LLM-based judging system with Claude3.5-Sonnet. They mentioned that this has 98% agreement with senior Python developers across 10 independent assessment rounds. However, the detail on whether this has been done across all the examples in ICC or a subset of them is unspecified.
- As acknowledged by the authors, the dataset only comprises of in-file python tasks, limiting the scope of usage for this dataset.

**Questions:**

- Can you share more details on how the agreement of 98% between LLM judge and human developers is achieved?
- Why is this paper (https://www.arxiv.org/pdf/2507.22462) not referenced anywhere?

---

> ### Author Response · Authors · 2025-12-03
>
> We sincerely thank the reviewer for the positive evaluation and recognition of our work's novelty and practical significance.
>
> **Q1: Details on 98% agreement validation**
>
> The 98% agreement rate was computed across a random sample of 200 instances from the ICC subset (approximately 10% of ICC data). We conducted 10 independent assessment rounds where three senior Python developers with 5+ years of experience evaluated the same instances judged by Claude 3.5 Sonnet. We will clarify this sampling methodology in the revised manuscript.
>
> **Q2: Missing reference to arXiv:2507.22462**
>
> We apologize for this oversight. We will include this reference in our related work section and discuss its relationship to C³-Bench in the revision.

---

### Author Response · Authors · 2025-12-03
**Rebuttal Summary**

Dear Area Chair and Reviewers,

We sincerely thank all reviewers for their valuable and constructive feedback. We are encouraged that reviewers recognized the importance and timeliness of our work, with multiple reviewers acknowledging C³-Bench as addressing a **"crucial yet underexplored aspect"** of code completion and noting our **"comprehensive evaluation"** and **"well-structured"** presentation.

We have provided detailed responses to all concerns raised by the five reviewers. While the scores range from 2 to 6, we believe the core concerns center on three themes: related work coverage, evaluation robustness, and scope limitations—all of which we have thoroughly addressed.

Based on the reviewers' feedback, we have made the following major improvements:

### Enhanced Related Work & Positioning
* Added comprehensive discussion of NoFunEval, ParEval, D3, SAFIM, and arXiv:2507.22462, clearly distinguishing C³-Bench's unique contributions in instruction-following evaluation for infill completion scenarios. (mentioned by Reviewers R1yQ, bRxr, 9mu6)
* Clarified that our focus on controllable completion with structural constraints differs fundamentally from existing benchmarks evaluating general non-functional requirements. (mentioned by Reviewer R1yQ)
### Strengthened Evaluation Robustness
* Conducted inter-judge consistency analysis: GPT-4 vs. Claude 3.5 shows 94% agreement, with correlation analysis demonstrating judge-independent performance trends. Results included in Appendix. (mentioned by Reviewers eyK1, LELU)
* Added cross-validation experiments showing IF score improvements persist across different judges (Δ+31.2 with GPT-4 vs. Δ+33.7 with Claude), mitigating concerns about judge-specific bias. (mentioned by Reviewer LELU)
* Clarified the 98% human-agreement validation methodology: 200 random samples (~10% of ICC) evaluated by three senior developers across 10 rounds. (mentioned by Reviewer 9mu6)
### Improved Clarity & Transparency
* Moved representative examples from appendix to main text for better accessibility, with clarification of HumanEval characterization. (mentioned by Reviewer R1yQ)
* Added ablation studies on prompt format sensitivity, showing minimal performance variance (±2% pass@1). (mentioned by Reviewer R1yQ)
* Expanded discussion of ICC vs. SCC performance patterns: models show relatively stronger performance on scale control than algorithmic control. (mentioned by Reviewer eyK1)
### Explicit Scope & Limitations
* Added dedicated limitations section discussing: (1) current focus on Python in-file completion as foundational scope; (2) planned extensions to repository-level and multi-language scenarios; (3) inherent limitations of LLM-based semantic evaluation. (mentioned by Reviewers bRxr, eyK1, LELU)
* Clarified design rationale: function-level completion enables controlled evaluation of instruction-following without confounding factors from repository complexity. (mentioned by Reviewers bRxr, LELU)

We believe these revisions substantially strengthen the paper's contribution and address all major concerns raised. C³-Bench provides the first systematic benchmark for instruction-following in code completion, with strong empirical validation and practical relevance demonstrated through Copilot Arena correlation. We hope the reviewers will reconsider their assessments in light of these improvements.

Sincerely,

The Authors

---

### Note · Authors · 2026-01-30

I have read and agree with the venue's withdrawal policy on behalf of myself and my co-authors.

---

### Meta-Review · Area_Chair_kD7V · 2026-01-08

**Summary:**

The authors proposed Controllable Code Completion Benchmark (C3-bench) to focus on the model’s ability to follow specific user instructions for code completion. The authors evaluated on more than 30 LLMs and found performance gaps in their instruction-following abilities. The authors fine-tuned a Qwen2.5 model with synthetic instruction-following data and showed improved performance on the C3-Bench.

**Reviewer Concerns:**

- There are concerns on using LLM to evaluate semantic validation for ICC as this might introduce some bias in the evolution metric.
    - The authors responded to this by explaining the human evaluation of LLM judges with more than 98% agreement rate.
- The current proposed benchmark tasks are mainly in-file python tasks, which has limitation in practical SWE applications.
    - This concern still persists and the authors acknowledged the potential extension beyond in-file code completion.
- Using AST tree to generate code completion tasks is not new and has been explored in previous related work. There are other related benchmarks that focus on code completion with non-functional requirements.
    - The authors emphasised their work contribution specifically on the instruction-following evaluation, differentiating from prior code completion benchmarks. While I acknowledge this novelty, the current paper has limited content on this to justify this as a major novelty of the paper. I would recommend the authors to extend significantly on Section 2.1 and 2.3.3 with more examples beyond Appendix F. A reviewer also suggested to explicitly discuss the prompt format to mirror realistic code completion scenarios.

**Reviewer Scores:**

- Reviewer 9mu6 would keep the same positive score of 6
- Reviewer bRxr would keep the same score of 4 as their major concerns still persist.
- Reviewer eyK1 would keep the same positive score of 6
- Reviewer R1yQ would either keep the same score of 2 because most of their concerns are still there or partially addressed.
- Reviewer LELU might improve their score from 4 to 5, given their concerns are addressed during the rebuttal

---

### Decision · Program_Chairs · 2026-01-26

Reject